# Hippo in Gastric Cancer: From Signalling to Therapy

**DOI:** 10.3390/cancers14092282

**Published:** 2022-05-03

**Authors:** Lornella Seeneevassen, Pierre Dubus, Caroline Gronnier, Christine Varon

**Affiliations:** 1BRIC (BoRdeaux Institute of onCology), UMR1312, INSERM, University of Bordeaux, F-33000 Bordeaux, France; lornella.seeneevassen@u-bordeaux.fr (L.S.); pierre.dubus@u-bordeaux.fr (P.D.); caroline.gronnier@chu-bordeaux.fr (C.G.); 2Department of Histology and Pathology, CHU Bordeaux, F-33000 Bordeaux, France; 3Department of Digestive Surgery, Haut-Lévêque Hospital, CHU Bordeaux, F-33000 Bordeaux, France

**Keywords:** gastric cancer, hippo, YAP, TAZ, cancer stem cells, CD44, cancer therapy, LIF, verteporfin

## Abstract

**Simple Summary:**

The Hippo signalling pathway is one of the most crucial and complex ones in physiology, and there is no doubt that the regulatory mechanisms it possesses are various. The role of this signalisation process in tissue homeostasis makes it keen to lead to cancerous processes when dysregulated. This review relates Hippo signalling and, more particularly, its role in gastric carcinogenesis and what has been attempted until today to encounter its disruption in this context. The Hippo pathway effectors YAP and TAZ are found to have a particularly important role in this disease. Different strategies, which can be used for their targeting in the GC context, are related in this review, may they be through their direct inhibition or through the activation of upstream Hippo kinases. Understanding the dysregulation of the organ homeostasis-regulating pathway in the cancer context is an important step towards the development of anti-gastric cancer therapeutical strategies.

**Abstract:**

The Hippo pathway is one of the most important ones in mammals. Its key functions in cell proliferation, tissue growth, repair, and homeostasis make it the most crucial one to be controlled. Many means have been deployed for its regulation, since this pathway is not only composed of core regulatory components, but it also communicates with and regulates various other pathways, making this signalisation even more complex. Its role in cancer has been studied more and more over the past few years, and it presents YAP/TAZ as the major oncogenic actors. In this review, we relate how vital this pathway is for different organs, and how regulatory mechanisms have been bypassed to lead to cancerous states. Most studies present an upregulation status of YAP/TAZ, and urge the need to target them. A focus is made here on gastric carcinogenesis, its main dysregulations, and the major strategies adopted and tested to counteract Hippo pathway disbalance in this disease. Hippo pathway targeting can be achieved by various means, which are described in this review. Many studies have tested different potential molecules, which are detailed hereby. Though not all tested in gastric cancer, they could represent a real interest.

## 1. The Hippo Pathway

### 1.1. Overview

The Salvador-Warts-Hippo pathway (SAV1-LATS1/2-MST1/2 in mammals) is the key regulator of organ size and tissue homeostasis in physiology. Ever since its discovery in *Drosophila* [1,2], described as “hippopotamus phenotype” (contributing to its name), where its deregulation induces a dramatic tissue overgrowth, this evolutionary conserved pathway has been described in divers physiological and pathological processes; from cell growth, proliferation, cell-cell contact, cell density and cell polarity control, stemness, shear stress, tissue homeostasis, and repair and regeneration, to cancer [3,4,5,6,7,8,9,10,11,12,13]. Genetic screenings for genes involved in cell growth led to the discovery of its different members [5,6,14]. This highly conserved pathway in mammals is made up of two central groups of elements driving its role: 1) a serine-threonine kinase core comprising of Mammalian Sterile-20 such as 1/2 (MST1/2) and Large tumour suppressor 1/2 (LATS1/2) kinases and, 2) a transcriptional module containing Yes-associated protein (YAP) [15,16,17] and/or a Transcriptional co-activator, with a PDZ binding motif (WWTR1 or TAZ) (Figure 1).

YAP/TAZ are transcriptional co-activators. They do not possess DNA-binding motifs and need DNA-binding partners to accomplish their roles. The major ones are TEA domain family member 1–4 (TEAD1-4), also known as the Transcriptional enhancer factor (TEF) in mammals, and Scalloped (Sd) in *Drosophila* [18,19]. TEAD fixation by YAP/TAZ is significant in the mediation of its biological functions. Screening of the human transcription factor library, coupled with luciferase transcription reporter assays and/or yap co-transfection, allowed the identification of TEADs as the major actor of YAP binding to gene promoters and of YAP-induced transcriptional activity [6,19].

Studies show that YAP/TAZ are the effectors carrying the functional activity of the pathway. Mutations of Yorkie (Yki), YAP/TAZ ortholog in *Drosophila*, lead to decrease in proliferation, while the gigantism phenotype and liver tissue expansion are observed with its overexpression, coinciding with the phenotype observed upon Hippo kinases mutation. YAP deletion in mice decreases cell overgrowth phenotype caused by a lack of *MST1/2* [5,8]. Active YAP mutant transgenic expression or dysregulation, through deletion of Hippo kinase members at the embryonic stage, leads to the hyperproliferation of cardiomyocytes and heart enlargement [20,21,22,23].

In addition, YAP1 overactivation in the intestines causes enlargement of the pool of multipotent undifferentiated progenitor cells, which undergo differentiation once YAP induction is ceased [8,11]. YAP inhibition also re-establishes cell-contact inhibition. Strict control of these effectors is, thus, required for proper tissue homeostasis.

### 1.2. Regulation Mechanisms

Hippo pathway regulators intervene at different levels for the maintenance of proper tissue homeostasis (Figure 1).

#### 1.2.1. YAP/TAZ Effectors Regulation

Hippo effectors are, indeed, tightly controlled by upstream members of the pathway, the hippo kinases, and their regulatory partners, Salvador Family WW Domain Containing Protein 1 (SAV1) and MOB Kinase Activator 1A/B (MOB1A/B), playing a role in the activation of MST1/2 and LATS1/2, respectively. Phosphorylation cascade of members of this kinase core induces the phosphorylation of YAP/TAZ effectors and, thus, represses their nuclear localisation and activity [3,6,9,18,24]. This is demonstrated in hippo kinases inhibition models, where nuclear accumulation of the effectors is observed [9,25].

Among the inhibition mechanisms are cytoplasm sequestration or nuclear exclusion mechanisms. YAP/TAZ are active in the nucleus and their nuclear accumulation is, thus, elemental for their function as co-transcription factors. Hippo kinase LATS1/2 phosphorylate YAP and TAZ on Ser-127 (p-YAP^SER127^) and Ser-89 (p-TAZ^SER89^) residues, respectively, allowing their binding to 14-3-3 proteins, thus retaining them in the cytoplasm [3,18]. In addition, 14-3-3 proteins knock-down (KD) induces nuclear accumulation, which is consistent with this observation.

Other 14-3-3 proteins-independent mechanisms also exist [25]. For example, LATS1/2 also phosphorylates YAP and TAZ on Ser-381 and Ser-311, respectively, modulating their protein stability, by inducing their polyubiquitination, and their addressing to the proteasome, for degradation. Indeed, studies have shown that TAZ Ser-311 phosphorylation by LATS1/2 can prime it to be further phosphorylated on Ser-314 on their phosphodegron motif by Casein kinase 1 (CK1), allowing interaction with β-TrCP, and, then, with SCF E3 ubiquitin ligases, to be directed to the proteasome for degradation [10,26]. This effector inhibition mechanism is absent in the *Drosophila* model.

Furthermore, YAP/TAZ contains a COOH-terminal domain, allowing interaction with proteins having PDZ binding domains [3,18,27]. This domain is important for YAP/TAZ regulation. Its absence causes cytoplasmic re-localisation and activity inhibition. YAP and TAZ are able to fixe Zonula occludens-2 protein (ZO-2) (and also ZO-1 for TAZ) and co-localise with it in the nucleus [28,29]. In addition, Vestigial-like 1-4 (VGLL1-4) contain Tondu domains (TDU), which have the capacity of binding TEAD. In so doing, VGLL4, for example, is able to inhibit YAP-TEAD1 interaction, and target TEAD1 for degradation [30]. Likewise, Runt-related transcription factor 3 (RUNX3) physically interacts with TEAD at its N-terminal region and, consequently, stops YAP from fixing to it and reduces its DNA-binding capacity as well as the downstream signalling [31].

YAP/TAZ-TEAD regulation is, therefore, most of the time achieved via the hippo kinase members, whether inhibiting their function or not.

#### 1.2.2. Hippo Kinase Core and Upstream Partners Regulation

The Hippo kinases are also very well regulated, allowing them to sense the need to exert their inhibitory effect on YAP and TAZ. Among the major regulators of the Hippo pathway are mechanical cues such as cell density, cell-cell adhesions, and apico-basal polarity [3,32]. Proteins involved in the maintenance of these cellular events are, thus, implicated in Hippo regulation.

Merlin, coded by neurofibromatosis gene *NF2*, is an essential regulator of Hippo kinases. It is localised at tight and adherens junctions, when cells are at high density and induce activation of core kinases, which in turn inhibit YAP/TAZ effectors to limit cell proliferation [33,34]. Merlin induces YAP phosphorylation by LATS by several means: it (1) binds with LATS, transporting it to the cell membrane and stimulating a complex formation with MST and SAV and its further activation; (2) promotes the assembly of scaffold proteins to facilitate LATS-YAP interaction and phosphorylation. Actin disruption can activate this Merlin-dependent process [35]. Merlin is critical in liver homeostasis and stem cell niche regulation [36].

Another important protein in the maintenance of cell and tissue integrity and, thus, having a role in the regulation of the Hippo pathway, is Scribble, an adaptor protein, which is important in cell polarity. It is localised at the cell membrane and, like Merlin, is able to assemble a complex, here made of MST, LATS, and TAZ, which is necessary for LATS activation and the consequent phosphorylation cascade [37].

Furthermore, apical proteins such as Crumbs and the Angiomotin family (AMOTs), as well as cadherins-actin cytoskeleton linker proteins such as α-Catenin, are involved in YAP/TAZ regulation by managing their localisation through phosphorylation [38,39,40,41].

Hippo pathway regulation can also imply cell surface receptors such as G-protein coupled receptors (GPCRs) [32] and the Leukaemia Inhibitory Factor receptor (LIFR) [42,43], part of the JAK/STAT signalling pathway. In addition, AMOTp130 isoform can also act as substrate for LATS1/2. AMOTp130-LATS1/2 signalling is inhibited by serum LPA, through activation of Gα_s_-coupled GPCR signalling. In this context, the LATS1/2 Ser-175 phosphorylation of AMOTp130 can disrupt the latter’s interaction with F-actin and decrease stress fibres and focal adhesions. In so doing, LATS1/2 and AMOTp130^SER175^ repress endothelial cell migration in vitro and angiogenesis in zebrafish models [44].

YAP/TAZ activation proteins also exist, for example, the Protein phosphatase 2A (PP2A) complex STRIPAK, which exerts a negative control on Hippo kinases through their de-phosphorylation, thus inducing YAP/TAZ [45].

#### 1.2.3. Hippo Pathway Crosstalk with/and Regulation by Other Signalling Pathways

The Hippo pathway is also able to interact with other cell signalling pathways and these crosstalks render signalisation even more complex. Some of these crosstalks are presented hereafter.

○AKT/p73 related pathway

p73, a p53 homologue able to induce the same anti-proliferative, pro-apoptotic, and pro-differentiation p53-like properties, is found to physically interact with YAP [46]. The PPPPY motif it contains can associate with the WW domain of YAP, and this interaction was described in H1299 human non-small cell lung cancer (NSCLC) cells, where YAP acts as a co-transcriptional activator of p73 [47]. Basu et al. show that both YAP and p73 are required for the induction of p73 target gene *BAX*, and that p73 acts downstream of YAP in human bone osteosarcoma epithelial cells (U20S) [48]. Furthermore, YAP-p73 signalling is regulated by AKT phosphorylation of YAP on Ser-127, inducing its cytoplasmic retention, and the 14-3-3 protein-interaction hypothesis was emitted as an explanation.

○JNK/AP1/c-Jun/p73 pathway

Danovi et al. demonstrate that YAP promoter contains several AP-1 binding sites, and that ectopic expression of an AP-1 family member, c-Jun, in U20S cells induces YAP mRNA expression in a c-Jun dependent manner [49]. Furthermore, they confirm that YAP is important and sufficient in c-Jun dependent p73 stabilization and is a critical effector of c-Jun-induced apoptosis. c-Jun N-terminal kinase (JNK), an upstream activator of c-Jun, might be involved in c-Jun-related YAP upregulation. Moreover, it has been shown in U20S, MCF-7 breast cancer, and BWT skin squamous cell carcinoma cells that JNK1/2 are able to directly phosphorylate YAP on five distinct sites; THR-119, SER-138, THR-153, SER-31, and THR362, the latter two being the most phosphorylated in vivo. In so doing, YAP acts as an apoptosis inductor [50].

○Src/c-Abl/p73 pathway

The Src family of a non-tyrosine receptor kinases member, c-Abl, is found to directly phosphorylate YAP on TYR-357 in response to DNA damage and stabilize it. YAP^TYR357^ has better affinity for p73 and better induces its pro-apoptotic effects in HEK293 cells [51].

○Wnt/β-catenin pathway

Azzolin et al. demonstrate that YAP/TAZ are components of the β-catenin destruction complex, having a role in β-catenin inactivation as well as in YAP/TAZ transcriptional inhibition through cytoplasmic sequestration [52]. YAP/TAZ interacts with the complex via Axin, as shown in HEK293 cells, and, in the absence of Wnt signalling, β-catenin is recruited to the complex, phosphorylated by glycogen synthase kinase-3 (GSK3) and degraded by β-TrCP ubiquitin ligase [52,53]. Docking of β-TrCP to the destruction complex requires YAP/TAZ. Wnt-activated signalling induces LRP6 accumulation, which competes with YAP/TAZ for Axin binding. Axin, thus releasing YAP/TAZ and dislodging it from the destruction complex, blocks β-TrCP recruitment. Wnt and YAP/TAZ are free to stimulate their respective target genes expression. Wnt/β-catenin and YAP/TAZ are, thus, closely related in their regulations [52,54].

YAP is also able to bind the SH3 domain of the Src family kinase c-Yes (also called YES1), through its SH3-binding motif [15]. YAP, transcription factor TBX5, and β-catenin form a complex that can be re-localised to anti-apoptotic gene promoters following YAP phosphorylation by YES1 [55]. The authors found YAP to be crucial in the proliferation of β-catenin active cells, while TAZ, also related to the Wnt pathway through its role in the inhibition of DVL1 [56], does not affect the cells proliferation when suppressed.

○TGFβ/BMP/Smad pathway

YAP/TAZ are also found to act as binding proteins for Smads, acting on downstream TGFβ and BMP signalling [57]. YAP interacts with the PY motif of Smad7, through its WW domains, and in so doing participates in Smad7-induced TGFβ inhibition [58]. Furthermore, YAP/TAZ can control the localization of Smad2/3 complexes after cell density-related stimuli. In conditions of high cell density, YAP/TAZ cytoplasmic localization, induced by the Hippo kinases and Crumbs elements, leads to the cytoplasmic sequestration of Smad2/3-4 and abrogates the TGFβ activation of Smad2/3-4 [59,60]. On the contrary, low cell density implies nuclear translocation of YAP/TAZ, TGFβ-dependent phosphorylation of Smad2/3-4, and nuclear translocation.

○PI3K/AKT/mTOR pathway

Communications between mTORC1 and Hippo pathways have also been demonstrated, which is not so surprising knowing that the mTORC1 pathway is one of the few other pathways having a crucial role in organ size in physiology. Tumaneng et al. showed that cells overexpressing YAP had an increased phosphorylation of S6K^THR389^ and AKT^SER473^, members of the PI3K/AKT pathway and direct substrates of mTORC1/2, while YAP KD cells presented a decrease in their phosphorylation, demonstrating the crosstalk between the two pathways [61]. Furthermore, Gan et al. show that LATS1/2 phosphorylate Raptor is on SER606 residue, thus altering its interaction with Rheb and attenuating mTORC1 activation [62]. They, thus, demonstrate a balance between mTOR and Hippo pathways, one being attenuated when the other is activated to maintain proper cellular and tissular homeostasis, since a deficiency in Raptor^SER606^ modifies cell growth, proliferation, and metabolism.

○Pax8/TFF-1 pathway

TAZ interacts with Paired box gene 8 (Pax8) and Thyroid transcription factor-1 (TFF-1), thus contributing to the differentiation of thyroid cells both in vitro and in vivo. The interaction involves the TAZ WW domain as well as the protein’s N-terminal and C-terminal domains [63]. TAZ binding to Pax8 and TFF-1 transcriptional factors induce their activity on the thyroglobulin promoter, responsible for the transcription of genes involved in thyroid development for example. TAZ binding to TFF-1 also plays a role in the lung formation and differentiation of respiratory epithelial cells [64].

○EGFR/RAF/MEK/ERK pathway

The Hippo pathway was also found to be interconnected to the MAK/ERK pathway. Indeed, ERK1/2 inhibition is found to decrease YAP/TEAD transcriptional activity and downstream target genes expression by promoting its degradation, while ERK2 overexpression promotes YAP expression, suggesting a regulation of YAP by ERK and its upstream regulator MEK [65]. Similarly, hippo effector YAP degradation and low activity is observed when ERK1/2 is suppressed by siRNA strategies, confirming the role of this RAF/MEK/ERK signalling in YAP control [65]. Moreover, YAP KO can decrease EGFR expression and its downstream ERK signalling, also demonstrating ERK pathway regulation by YAP [66].

○AMPK pathway

The AMPK pathway is also found to interact with the Hippo pathway and participate in its regulation [67]. AMP-activated protein kinase (AMPK) phosphorylates AMOTL1 on its SER793 residue, leading to its stabilisation, which induces YAP inhibition.

○EMT/ZEB1/SNAIL/SLUG pathway

YAP is found to interact directly with ZEB1, turning the latter into a transcriptional co-activator of a set of ZEB1-YAP target genes. Both proteins are able to bind to *CTGF*, *CYR61*, *SDPR*, and *AXL* promoters, and simultaneous interaction is observed, at least for *CTGF* and *CYR61* [68]. TEAD binding sites are necessary for these co-activation capacities. This interaction was shown not to exist for TAZ. Nevertheless, TAZ has been described as able to control *ZEB1* transcription, by binding to its promoter in retinal pigment epithelium (RPE) cells, causing their proliferation and dedifferentiation [69].

In addition, YAP/TAZ are able to interact and complex with SNAIL/SLUG and promote transcriptional activity of TEAD, inducing the transcription of *CTGF*, *ANKRD1*, *AXL*, and *DDK1* involved in cell proliferation [70]. TAZ and not YAP is able to complexify with SNAIL/SLUG to promote RUNX2 transcriptional activity and the expression of *BGLAP2*, *OSTERIX*, and *ALP* having a role in differentiation. SNAIL/SLUG binding domains on YAP/TEAD are found to be their WW domains [70].

A proper balance between positive and negative signalling (Figure 2) seems necessary for the control of this essential cell signalling pathway and the maintenance of tissue integrity.

### 1.3. Hippo and Cancer

This pathway’s implication in cellular and tissular vital processes makes its dysregulation highly critical. Despite the numerous levels of regulation, the hippo pathway is found to be involved in various cancers. In *Drosophila*, mutations altering the function of Hippo kinases induce hyperproliferation and decreased apoptosis, leading to the appearance of tumours. This first demonstrated the tumour suppressor characteristics of Hippo kinases [3,12]. The liver tissue enlargement observed after mutation of Hippo kinases can be attributed to uncontrolled cell growth observed in cancers [5,8]. Furthermore, activation of YAP drosophila homologue *Yki* led to unrestrained growth of cells, which is one prerequisite for tumour formation [5].

Studies relate the role of YAP/TAZ hyperactivation in many cancers. The critical genes regulated by these transcriptional co-activators make it easy to go over to the dark side when disturbed, tending towards some cancer hallmarks such as sustaining proliferative signalling, resisting cell death, activating invasion and metastasis, and tumour-promoting inflammation, among others [71,72]. Though it has high functional potential, activating mutations of *YAP* gene are not described in cancer. Nonetheless, it is found to be localised on the 11q22 amplicon, amplified in various cancers, which could explain its hyperactivity [73]. Furthermore, inactive mutations or epigenetic regulations of Hippo kinases or close members are often observed in tumours [73].

Plouffe et al. show that YAP and its paralogue TAZ have distinct roles apart from their overlapping functions [74]. Hippo target genes induction can also differ depending on the effector involved. For example, *CTGF* and *CYR61* are regulated by both YAP and TAZ, since serum-induced stimulation of these genes is decreased in YAP/TAZ-KO cells. Furthermore, YAP deletion seems to have a greater effect than TAZ deletion on *CTGF* and *CYR61* expression, though to a lower extent for the latter. This was demonstrated in three different cell lines, including HeLa cervical cancer, MCF7 breast cancer, and HEK293A human embryonic kidney cells. YAP KO, TAZ KO, and YAP/TAZ KO caused the induction of *LGR5*, showing the role of both effectors in *LGR5* repression. YAP and YAP/TAZ KO demonstrated the role of YAP in cell spreading, size, granularity, glucose uptake, proliferation, and migration, which decreased after its deletion compared to wild-type (WT) cells that resembled TAZ KO cells [74]. Interestingly, LATS1/2 KO cells gave the opposite of what was observed in YAP KO and YAP/TAZ KO cells. Nevertheless, this study was carried out on HEK293A human embryonic kidney cells in which, despite the higher TAZ mRNA expression than that of YAP, the highly dynamic regulation of TAZ made its protein less than twice that expressed by the YAP protein [74]. This could explain why YAP deletion affects more cell size and physiology than TAZ deletion. YAP different regulation of *CTGF* is, nonetheless, a solid fact, and the other YAP-dependent target genes identified were AMOTL2 and Fos-like antigen 1 (FOSL1).

Colorectal (CRC), NSCLC, and breast cancers, as well as hepatocellular carcinomas (HCC) and melanomas, for example, present high expression of YAP or TAZ. In CRC, circular RNA circPPP1R22A induces YAP activation, causing tumour growth and metastasis, which decreases in presence of peptide 17, an inhibitor of YAP [75].

In breast-cancer-induced bone marrow metastasis, hypoxic environments influence HIF-1α interaction with TAZ. Nuclear HIF-1α is associated with the epithelial-to-mesenchymal transition (EMT) process and interaction with Hippo effectors, and is negatively regulated by LATS1/2 [76,77]. YAP interaction with ZEB1, as discussed before, contributes to the expression of target gene having a role in poor survival of patients, therapy resistance, and increased metastasis in breast cancer [68].

Furthermore, breast cancer bone marrow metastasis can also be induced by crosstalk between ROR1-Her3 and Hippo-YAP, through inactivation of MST1 [78]. Indeed, the phosphorylation of HER3 at Tyr1307 by the ROR1 tyrosine kinase receptor induces the methylation of MST1 at Lys59 and its deactivation. *MOB1a/b* deletion causes breast [79] and lung tumours [80]. Furthermore, it has been demonstrated that breast cancer cells are able to promote YAP/TAZ expression and activity in cancer associated fibroblasts (CAFS), one of the components of the tumour micro-environment. This has, as a consequence, the remodelling and stiffening of extracellular matrix, related to CAFs’ pro-tumorigenic roles, as well as the angiogenesis improvement required for tumour growth [81].

YAP and TAZ overexpression in NSCLC are related to tumour development, progression, and a patient’s poor prognosis. Hyperactivation mutations of YAP are observed in this type of cancer, as is the downregulation of LATS2 in 60% of cases [82]. miR-135 is found to be highly expressed in NSCLC and is associated with poor survival outcomes. This microRNA is capable of increasing invasive and migration properties of cancer cells in vitro and metastasis in vivo, through the targeting of hippo kinase core members [83]. Moreover, YAP is able to activate EMT transcription factor Slug, which in turn inhibits the *BMF* pro-apoptotic factor. Cells enter a senescence-like dormant state and counter the drug-induced apoptosis [84]. RAF/MEK/ERK is found to contribute to NSCLC through YAP modulation, since the anti-ERK1/2 siRNA strategy discussed earlier led to a decrease in migration and an invasion of NSCLC cells, along with the decrease in YAP protein expression observed [65].

In the liver, inducible YAP expression as well as deletion of *NF2*, *SAV1*, or *MST1/2* leads to hepatomegaly and, further, to liver tumours [8,9,36,85]. HCC is characterised by the high expression of miRNA-665 and miRNA-3910, which have inhibitory effects on hippo kinases such as MST1, leading to decreased apoptosis and increased cell proliferation, migration, invasion, and EMT [18,86,87,88]. On the contrary, miRNA-195, having a tumour suppressor role, is decreased in HCC, and this is associated with the low survival rate of patients [89,90]. YAP activation is an early event in HCC, with the PDZ-binding domain being crucial for the activation of the cell proliferation gene *CTGF* [12]. In this type of cancer, CREB can promote transcriptional activity of YAP [91,92]. In addition, MEK1-YAP1 interaction leads to increased cell proliferation and maintenance of transformed neoplastic phenotype [93].

In melanomas, YAP is sufficient and necessary for invasion of cells and appearance of spontaneous metastasis [94]. Target genes involved in this phenomenon are *AXL*, *THBS1*, and *CYR61*. In skin cancers, deletion of *MST1/*2, surprisingly, does not have any flagrant effect, though YAP activation is implicated in the keratinocytes’ hyper-proliferation and the squamous cell carcinoma that is overcome, following the deletion of the *CTNNA1* encoding α-catenin [40]. Furthermore, methylation of the LATS1/2 promoter contributes to oral squamous cell carcinoma (OSCC) [95,96]. In addition, in pancreatic cancer, YAP1 cooperates with K-RAS and induces tumour survival and EMT, which is involved in cancer cell metastasis [97].

Β-catenin-YAP cooperation, as explained above, is important in the tumorigenicity of β-catenin active cells, namely SW480, SNU-C1, HCT116 colon cancer cells, AGS GC cells, and A549 lung cancer cells, among others [55]. Interestingly, conversely to what expected, neither β-catenin partner TCF2 nor YAP partner TEAD was implicated in this pro-tumorigenic effect [52,56]. TBX5 transcription factor is the one mediating the pro-proliferation signal by inducing the transcription of *BCL2L1* and *BIRC5*, and YES1 is essential in this process.

The Hippo pathway also plays an important role in the resistance to therapy of many cancers, such as gliomas, retinoblastomas, endometrial, bladder cell, pancreatic, and ovarian cancers [98,99,100,101,102,103]. YAP-TEAD is involved in NSCLC cells’ escape to Epidermal growth factor receptor (EGFR) tyrosine kinase inhibitor (TKI) treatment [84]. Indeed, resistant cells are found to have high expression and activity of YAP/TEAD.

Another dysregulation observed in cancers and linked to the Hippo pathway is EMT, which involves a loss of epithelial architecture. This affects Hippo membrane regulators such as Scribble, which gets delocalised, thus activating YAP/TAZ and contributing to tumorigenesis and CSCs [37]. This has also been demonstrated in breast cancer cells lacking E-cadherin, having a role in epithelium integrity, and where the Scribble delocalisation phenotype can be counteracted by Hippo kinases reactivation [42]. Indeed, a perturbed E-cadherin/α-Catenin complex leads to decreased YAP phosphorylation and induced cancer-related transcriptional activity.

The role of the Hippo pathway in tissue homeostasis is greatly responsible for tumour appearance and maintenance when dysregulated. Cancer is a complex disease, and YAP/TAZ oncogenic ability seems to be involved in diverse parts of this ailment, from tumour initiation to tumour-immune cells’ crosstalk, extracellular matrix remodelling, and dissemination [3,7,12,104,105]. Disbalance of Hippo-YAP/TAZ regulation is at the root of many cancers, amongst which is gastric cancer (GC), on which we will focus in the next part of this review.

## 2. Hippo Pathway in Gastric Carcinogenesis

### 2.1. Hippo Pathway in Helicobacter-Mediated Gastric Carcinogenesis

GC is a major health concern, and the Hippo pathway is implicated, since the very beginning of gastric carcinogenesis induced by chronic infection with *Helicobacter pylori* (*H. pylori*) is classified as a type 1 carcinogen and the principal cause of GC [106,107,108,109].

MST1/2 and LATS1/2 Hippo kinases are found downregulated, and YAP/TAZ effectors upregulated, in GC [110,111,112] (Figure 3). Nuclear expression of YAP/TAZ, related to their activity, is associated with a poor prognosis in GC patients, particularly in those with intestinal-type GC for YAP [32,42,113,114,115,116]. YAP is found overactivated in gastric carcinogenesis, and *H. pylori* infection stimulates both Hippo downstream effectors YAP/TAZ, as demonstrated by transcriptomic analyses in gastric epithelial cells after *H. pylori* infection [106]. This effect is abrogated by infection with CagA-mutant strains, showing the role of *H. pylori* in YAP hyperactivation and, most particularly, of the bacterium’s CagA oncoprotein.

Furthermore, YAP inhibition affected *H. pylori*-induced EMT, demonstrating the pathway’s contribution to *H. pylori*-related gastric carcinogenesis. Molina et al. have shown that *H. pylori* infection of gastric epithelial cells induces the expression of both YAP1 and LATS2 from the gastritis stage (early stage of the Correa’s cascade of *H. pylori*-induced gastric carcinogenesis) [43,106], which continues increasing at the intestinal metaplasia and GC stages [106]. YAP1-TEAD pro-proliferation and pro-survival-related genes were found to be increased after infection. Time-course infection with *H. pylori* showed that YAP1 is activated as early as 2h after the bacterial infection and causes an increase in target genes, including LATS2 acting as a regulatory feedback loop, thus controlling *H. pylori*-induced YAP/TAZ activation and cell growth from 5 to 24 h post-infection. An equilibrium, thus, exists for the maintenance of YAP1 proper balance as well as proper epithelial cell differentiation and survival in response to *H. pylori* infection, to limit epithelial cell identity loss [106].

Furthermore, nuclear YAP/TAZ-TEAD activity is induced following *H. pylori* infection of GC cells. TAZ is co-expressed and found co-localised with EMT-related transcriptional factor ZEB1 in cells having a mesenchymal phenotype, and at the invasive fronts of gastric tumours. TAZ-ZEB1 cooperation in *H. pylori*-induced GC is demonstrated by TAZ silencing, which decreases ZEB1 expression and EMT, as well as invasion and cancer stem cell (CSC) tumoursphere formation properties acquired by GC cells after *H. pylori* infection [107]. This could be related to direct TAZ interaction with the promoter of *ZEB1*, as described in RPE cells [69].

### 2.2. Hippo Pathway in GC and CSCs

CSCs’ presence in tumours is a major problem in GC, like in many other solid tumours [108,118,119,120,121,122,123]. CSCs correspond to a subpopulation of cells within tumours at the origin of tumour initiation, growth, dissemination, and resistance to current treatments. They have asymmetrical division properties, allowing their self-renewal and parallel proliferation and differentiation, at the base of tumour heterogeneity. Bessède et al. reported that, during the gastric carcinogenesis cascade, *H. pylori* infection induces the emergence of CD44+ cells carrying CSC properties, through the EMT process [124]. Giraud et al. recently demonstrated through transcriptomics analysis that CD44+ gastric CSCs have a Hippo-pathway-rich signature, with an overexpression of effectors and target genes such as *AREG, CCND1, CDX2, CYR61, BIRC5, ID1, IGFBP3, JAG1, LATS2, SMAD7,* and *MYC* [125]. Interestingly, *TEAD1/4* were also found upregulated in CD44+ cells compared to CD44-non-CSCs, while TEAD inhibitors, *VGLL4* and *RUNX3*, were downregulated. Furthermore, the authors also demonstrated that residual chemotherapy-treated cells, which are the most resistant ones, present higher expression of *YAP1*, *TAZ*, *CYR61*, and *CTGF*, all associated to high YAP/TAZ-TEAD activity, compared to non-treated cells.

Fujimoto et al. have shown that Protease-activated receptor 1 (PAR1) stimulation sustains cells with stem-like properties, and induces cell invasion and EMT through inhibition of LATS via Rho A GTPase [126]. PAR1 induces Rho A-induced inhibition of LATS1/2 phosphorylation, leading to activated YAP. Another receptor, Peroxisome proliferator-activated receptor δ (PPAR δ), is found to interact with YAP1, inducing the transcription and upregulation of SOX9, a gastrointestinal stem cell marker, in premalignant lesions [127]. SOX9 induces CSC properties in non-cancerous cells [128], promotes invasion and metastasis, and is overexpressed in patients at advanced stages of GC [11].

Gastric carcinogenesis process was also studied independently of *H. pylori* infection. Overexpression of hippo effectors, through *LATS1/2* knockout (KO) in mice pyloric stem cells, generated transformation of mice normal gastric epithelium to low-grade intraepithelial neoplasia, followed by intramucosal carcinoma [129]. YAP/TAZ were found to initiate GC via MYC as a downstream mediator and direct transcriptional target of YAP [129]. Inhibition of MYC interfered with YAP/TAZ-induced carcinogenesis.

Another study showed LATS1 low expression in GC patients, and this was correlated with lymph node metastasis, poor prognosis, and tumour relapse [130]. LATS1 is found to decrease GC cell proliferation and invasion in vitro, and tumour growth and metastasis in vivo, by inhibiting YAP nuclear translocation and, thus, expression of pro-proliferative and pro-invasive target genes, among which are *YAP*, *CTGF*, *PCNA*, *MMP-2*, and *MMP-9*. TAZ overexpression, particularly in diffuse type GC, is associated with EMT profile and low survival of patients [43,125,131].

RUNX3 has been shown to be inactivated in about 60% GC. Subsequently, the tumour suppressor effect of this TEAD interactor on the YAP-TEAD complex is affected in GC, and hyperactivation is noted [31]. The RUNX3 mutation in GC impairs its activity towards a pro-YAP signalling phenotype. Likewise, RUNX2 is implicated in gastric carcinogenesis through the Hippo pathway. It is highly expressed in GC early stages and predicts poor prognosis [132,133]. Ectopic RUNX2 expression is implicated in GC invasion and metastasis process through binding to CXCR4 promoter, thus inducing the expression of CXCR4 [132]. It also induces tumoursphere formation and tumour initiation, showing an effect on the gastric CSC population [132]. CXCR4′s role is also described in CSC and metastasis in GC, as well as in pancreatic, breast, and colorectal cancer [134,135,136,137,138].

Zhou et al. show the role of methyltransferase 3 (METTL3), a major component of the m6A methyltransferase complex, in gastric carcinogenesis [139]. It is highly expressed in GC cases versus non-tumoral tissues. *METTL3* overexpression and KD show its role in GC cell proliferation, migration, and invasion in vitro and tumour growth in vivo. Interestingly, METTL3 positively controls YAP1 expression, which could explain its effects [139].

Moreover, Angiomotin Like 1 (AMOTL1) has been shown to interact with YAP1 and promote its nuclear translocation and its activity, thus contributing to gastric carcinogenesis [140]. The hippo pathway is also involved in Fibroblast growth factor receptor type 2 (FGFR2) gastric oncogenesis, through indirect YAP1 activation via the MAPK-c-Jun pathway [141]. Overexpression of transcriptional core TEAD1/4 and YAP/TAZ targets *CYR61* and *CTGF* is related to metastasis and bad prognosis. Furthermore, Tang et al. demonstrate a mechanism by which MST1/2 is turned off in some GC cases. Indeed, STRN3 was discovered as a PP2A regulatory subunit and was found to be able to recruit MST1/2, thus promoting its dephosphorylation and the consequent hippo effectors hyperactivation. dSTRIPAK forms part of the STRN3 molecule family, whose expression in induced in GC and associated with YAP activation and bad prognosis [142].

Additionally, microRNAs are also seen to intervene in the hippo pathway’s dysregulation in GC. MiR-125a-5p upregulates TAZ, TEAD2, and, therefore, their target genes, and stimulates cell survival, EMT, invasion, and tumour growth [87]. MiR-125a-5p is found in high levels at late stages of GC and co-expresses with TAZ and TEAD2. Similarly, miR-375 modulation is implicated in GC. Promoter methylation and histone deacetylation represses miR-375 expression in this disease, and this is associated with bad prognosis and lymph node metastasis [143]. Kang et al. show that miR-375 directly targets YAP1, TEAD, and CTGF, and that YAP1 re-expression partly stalls miR-375 tumour-suppressive effects. Furthermore, CTGF-KD demonstrates the same effects as YAP1-silencing and miR-375 ectopic expression on gastric tumours, demonstrating the role of Hippo effectors and their targets in GC. MiR-372 and MiR-373 are found to be implicated in GC cells AGS proliferation through inhibition of LATS2 gene expression [144].

Moreover, long noncoding RNAs (lncRNAs) have been demonstrated as having a role in hippo-pathway-induced gastric carcinogenesis. LncRNA *RP11-323N12.5* is the most highly expressed lncRNA in GC, according to the TCGA database. *RP11-323N12.5* is found to be correlated to YAP1 expression and to regulate it by acting on its promoter [145]. In so doing, it exerts pro-tumoral and pro-immunosuppressive functions. It is also regulated by YAP1/TAZ-TEAD transcription complex.

Besides, lncRNA *acv*3UTR is also found to be upregulated in GC. Ectopic expression of *acv*3UTR in GC cells demonstrated its role in cell growth promotion [146], through its negative effect on tumour suppressor miR-590-5p. Its expression correlates with that of YAP1. In addition, lncRNA *LINC00649*, enriched in GC, is found to induce cell proliferation, migration, and EMT in vitro, as well as tumorigenesis in vivo [147]. It is able to act on miR-16-5p, which is connected to YAP1 mRNA at the 3′UTR region. By doing so, it blocks the action of hippo kinases, which no longer phosphorylate YAP, and contributes to the expression of its pro-tumoral target genes [147]. On the contrary, Sun et al. identified lncRNA *LATS2-AS1-001*, which has an anti-tumoral effect in GC, through binding to the Enhancer of zeste homolog 2 (EZH2). This leads to LATS2 upregulation and YAP1 phosphorylation, thus decreasing cell viability, migration, and invasion. Apparently, this lncRNA is under-expressed in GC, which disables its effects [148].

Studies also show a relationship between cell metabolism and hippo-induced gastric carcinogenesis. Indeed, Liu et al. have shown the contribution of a nucleotide sugar transporter Solute carrier family, 35 B4 (SLC35B4), as having a crucial role in macromolecule glycosylation in GC [149]. The YAP1-TEAD complex directly activates SLC35B4, which induces cell proliferation in vitro and in vivo. It is more expressed in cancerous tissues compared to non-cancerous ones and is related to poor prognosis through YAP1′s action. In addition, Acylglycerol kinase (AGK), a lipid kinase involved in the production of physiological lysophosphatidic acid (LPA) and phosphatidic acid (PA), has been recognised for its pro-tumoral role when highly expressed [150]. It induces proliferation, invasion, and EMT phenotype in GC cells, effects that were abrogated by its KD. YAP1 plays a positive role on its regulation through binding to TEAD and to AGK’s promoter. In turn, in a positive feedback loop manner on YAP-TEAD, AGK inhibits hippo kinases, thus promoting YAP nuclear translocation and activity [150].

### 2.3. Hippo Pathway in GC Resistance to Therapy

The Hippo pathway has also been shown to have a role in therapy resistance in GC, like in many other cancers. There is, unfortunately, no targeted therapy in GC, except for Trastuzumab, which is approved for the treatment of HER2+ metastatic GCs. However, resistance mechanisms have been discovered that are also found at this level, rendering the search for other targets even more vital. Trastuzumab-resistant cells are found to have an overexpression of HER4, phosphorylated-HER4 (p-HER4), and vimentin (VIM), a mesenchymal phenotype marker, and a decrease in epithelial markers. Shi et al. demonstrate that in these cells, HER4 acts on its target YAP1 and induce the expression of genes involved in EMT and proliferation, leading to higher cells migration and growth as well as a decrease in HER2 and E-cadherin expression, leading to HER2-therapy resistance and contributing to mesenchymal phenotype acquisition [151]. Moreover, YAP stimulates GC cell proliferation, while its KD enhances cells’ sensitivity to cisplatin, showing the role of hippo effectors in GC chemoresistance [66]. In this context, the authors showed that this effect passes through regulation of EGFR/AKT/Extracellular signal-regulated kinase ½ (ERK1/2) by YAP. TAZ overexpression, particularly in diffuse type GC, is associated with the EMT profile, resistance to chemotherapy, and low survival of patients [152].

Interestingly, studies show that YAP and TAZ are not always expressed concomitantly in GC cells [106,107,153]. TAZ is found to be more expressed in gastric signet ring cells carcinomas (SRCC), a poorly differentiated type of GC, usually attributed to Lauren-diffuse-type GC. This type of GC is of a particularly bad prognosis, and this could be related to its high TAZ expression, which is also found to be higher in other poorly differentiated tumours compared to well-differentiated ones such as the Lauren intestinal type. Tiffon et al. demonstrated that in the SRCC GC cell line MKN45 and poorly differentiated GC patient-derived xenograft cells, TAZ is overexpressed and overactivated, controlling the expression of TAZ-TEAD target genes, including *CYR61* and *AXL* among others, and of the EMT main transcription factor ZEB1, which decrease in TAZ KD cells. TAZ is also related to GC cells’ EMT phenotype and invasiveness in this context [107]. Diffuse-type GC can also be characterised by low KRT17 intermediate filaments expression. This decreased KRT17 expression causes E-cadherin loss, EMT phenotype arousal, and metastasis in GC cells. Indeed, loss of intermediate filaments promotes cytoskeleton remodelling and reorganisation, which activates YAP and induces IL6 expression linked to increased metastasis [154].

The Hippo pathway and, most particularly, YAP and TAZ oncoproteins (Table 1), have a crucial role in gastric carcinogenesis, so therapeutic strategies involving their targeting could be of real efficiency in this poor prognosis, high relapse disease.

## 3. Anti-GC Strategies Involving the Hippo Pathway

Hippo pathway targeting can be achieved by either altering effectors’ oncogenic function or by reinforcing kinases’ tumour-suppressive role. Studies have explored the potential of different pre-existing molecules in GC (Table 2).

### 3.1. Downstream Strategies: Targetting Oncogenic YAP/TAZ-TEAD Signaling

YAP/TAZ-TEAD interaction is crucial for the function of the Hippo pathway effectors. Studies have shown the vital role of TEAD nuclear localization in this partnership. When TEAD is not localized in the nucleus, mobilizable unphosphorylated YAP is not able to translocate in the nucleus [74]. Altering this interaction could be the key against overactivation of this complex in cancer. Several strategies have been tested, and some are presented thereafter.

*RUNX family*: Member of the RUNX family RUNX3, which has a tumour suppressor role in GC like many other cancers [155] and is downregulated in cancers, can be used as a therapeutical strategy. Its action mechanism is its ability to associate with LATS1/2-phosphorylated YAP and facilitate its dissociation from the TEAD transcription factor [156]. The YAP-RUNX3 complex leads to decreased GC tumorigenesis, making its use as therapy interesting. Ectopic RUNX3 could be used to compensate its defects in GC [31] and as a competitor for TEAD binding, resulting in tumour-regressing effects. Indeed, RUNX3 overexpression inhibits colony formation of AGS and MKN28 cells in vitro and tumour growth in vivo, compared to the overexpression of mutated RUNX3_L121H_ impairing RUNX3-TEAD interaction, which had no effect on GC cells.

*VGLL4*: Mutations of TEAD residues have shown their role and importance in tumorigenesis and their interest as targets for therapy [157]. The TDU domain, as described before, is important for binding TEADs and VGLL4 and has been shown as a TDU-domain-containing protein, having tumour suppressive roles by destabilizing YAP-TEAD interaction [30,116]. Overexpression of *VGLL4* in GC cells induces apoptosis and decreased cell viability in vitro, while deletion of the TDU domain impaired VGLL4′s effect. Indeed, VGLL4 is in direct competition with YAP for TEAD fixation, and TDU domain interaction is necessary and sufficient to inhibit YAP-TEAD interaction and activity [116].

*Super-TDU*: Another strategy using endogenous-protein mechanisms is that of the super-TDU. Based on this, Jiao et al. designed an inhibitor peptide named “Super-TDU”, mimicking VGLL4 anti-tumoral activity in vitro and in vivo [116]. Co-immunoprecipitation experiments show that Super-TDU is able to decrease endogenous YAP-TEAD interaction. Moreover, it is able to decrease YAP-TEAD target genes expression such as *CYR61*, *CTGF*, *CDX2* as well as cell viability as well as colony formation of GC cells in vitro, compared to mutated Super-TDU. In addition, Super-TDU also impairs tumour growth in vivo [116]. This demonstrated how YAP-TEAD pro-tumoral signalling can depend on a short peptide, and how this could be used as an anti-GC strategy.

*Verteporfin*: Consistent with the previous observations that YAP/TAZ-TEAD interaction is crucial for the hippo effectors’ activity and function, Verteporfin (VP) strategy works by disrupting this interaction. VP is a benzoporphyrin derivate, FDA-approved for the photodynamic therapy of ocular diseases (Vysudine) such as age-related macular degeneration, pathologic myopia, and presumed ocular histoplasmosis, where it acts as a photosensitiser through ROS generation, leading to anti-angiogenic effects [158]. Moreover, VP competes with YAP and inhibits TEAD-YAP binding, thus limiting YAP-induced liver overgrowth in a mouse model [159]. This potential makes VP drug repositioning of real interest in YAP-TEAD targeting in GC.

VP inhibits GC cell growth and expression of migration-associated genes and oncogenes [160]. VP is found to affect FAT1 adhesion molecule, having a role in GC migration and invasion and reflect FAT1 silencing effects.

Hasegawa et al. show that cells have distinct sensitivities to VP, according to their YAP and TAZ expression levels [153]. The authors demonstrate that MKN45 cells, a poorly differentiated cell line having a TAZ-dominant expression, is more sensitive to VP than the MKN74 cell line, moderately differentiated with a YAP-dominant expression profile. Nevertheless, both cell lines responded to VP treatment by inducing YAP/TAZ cytoplasmic re-localisation, degradation, and a decrease in Survivin expression, an anti-apoptotic protein [153].

Wang et al. demonstrate that VP-induced YAP-TEAD disruption capacity could pass through its ability to stimulate the expression of 14-3-3 family protein, 14-3-3σ, which in turn promotes YAP nuclear-cytoplasmic translocation [169]. This effect requires p53 and determines the fact that VP treatment solution could better work in patients having p53 expression, allowing to propose more targeted options and limit treatment non-responsiveness.

Furthermore, Giraud et al. have demonstrated the role of VP in the GC CSC context. VP decreases the pool of CD44+ ALDH+ CSCs and the expression of several CSC markers in GC, thereby affecting GC cells tumoursphere-forming capacity and proliferation in vitro as well as tumour initiation and growth in PDX models in vivo. VP also affected the chemoresistance properties of gastric CSCs, showing its importance in complement of current chemotherapies in GC treatment [125].

VP can also be used as photodynamic therapy (PDT) in GC, since it results in apoptosis of the treated gastric cell lines. This PDT implies the induction of photochemical reactions to destroy tumour cells through singlet oxygen production [170]. This implies a possible use of VP both for CSC targeting and as a GC PDT strategy.

*Peptide 17*: Peptide 17 is able to inhibit YAP-TEAD interaction [171]. It was shown to inhibit lung cancer cells proliferation, apoptosis as well as colony formation, through its inhibitory effect on YAP-TEAD [172]. In addition, Peptide 17 is able to repress N6-methyladenosine (m6A)’s methyltransferase 3 (METTL3) expression, leading to a decrease in YAP1 mRNA expression and of the consequent tumour-promoting effects in GC [139]. The authors show that in AGS and MKN45, poorly differentiated cell lines overexpressing METTL3 compared to non-tumoral gastric mucosal epithelial cells GES-1, METTL3 increases the expression of YAP1 gene. YAP1 inhibition by peptide 17 can abrogate METTTL3-induced effects and, thus, depend on YAP-TEAD interaction, which can be targeted by peptide 17 [139].

*WZ35, a Curcumin analogue*: Chen et al. demonstrate the anti-GC effects of WZ35, exceeding those of its analogue Curcumin. Both decrease cell survival and proliferation, though to a higher extent for WZ35 [161]. It plays on cell glycolysis, increases the cellular ROS level, and presents anti-tumour effects in vivo. Most importantly, ROS represses YAP and activates Jun kinase (JUNK), which gets phosphorylated and promotes cell apoptosis. The same effects were demonstrated in breast cancer cells and NSCLC [161,162,173]. However, it is worth noting that in breast cancer, the authors show that ROS activates YAP and JNK, and, here, YAP acts as a tumour suppressor in a ROS-stimulating context, such as in lung squamous cell carcinoma [174]. Indeed, this dual effect of YAP could be related to its activating partners. For example, p73-related signalling confers pro-apoptotic properties, while being pro-growth for TEAD-associated signalling. Effect of curcumin/WZ35 was not attributed to the level of differentiation but different sensitivities of GC cells to treatment, according to their subtypes and to YAP and/or TAZ expression levels, as discussed above, which makes this an important aspect to treat in GC drug research.

### 3.2. Upstream Strategies: Hippo Kinases and/or Side-Pathways Stimulation

Apart from targeting YAP/TAZ-TEAD interactions and, thus, the functional effector core, another treatment strategy could be the modulation of the kinase core and upstream members of the hippo pathway in such a way as to stimulate YAP/TAZ phosphorylation and inhibition of the downstream effectors. Most of the strategies involving upper members of the pathway imply stimulation and crosstalk with other signalling pathways.

*Dobutamine*: Dobutamine is a β1-adrenoceptor agonist capable of stimulating β2 and α1-adrenoceptors too. It is used for treating patients with congestive heart failure and in dobutamine stress cardiovascular magnetic resonance, a test used in this domain [175,176,177]. Bao et al. demonstrated, thanks to a cell-based assay they developed, that dobutamine is capable of recruiting YAP to the cytoplasm, thus diminishing its nuclear translocation and activation of its target genes [178]. Its action passes through Protein kinase A (PKA) signalling and LATS1/2 activation. In combination with cisplatin, it inhibits GC tumour cell growth, migration, invasion, and CSC properties [163]. It promotes GC cell death by apoptosis and causes YAP to be more phosphorylated and cytoplasmic.

*Compound F10*: One Hippo kinase-activation strategy is cytoskeletal remodelling, an important aspect of hippo pathway regulation. Song et al. present a new tertiary amide derivative, containing benzothiazole motifs, named Compound F10, and having an anti-proliferative capacity on colorectal cancer cells HCT-116, GC cells MGC-803 and prostate cancer cells PC-3 [164]. Compound F10 represses tubulin polymerization and activates MST1/2 and the downstream signalling cascade, leading to the phosphorylation and inhibition of YAP. Consequently, expression of c-Myc and Bcl-2 is affected, and cells undergo apoptosis, less proliferation, and a decrease in sphere-forming capacity linked to CSC properties [164].

*Statins*: Lovastatin and Fluvastatin were also identified as YAP/TAZ inhibitors by Oku et al. Fluvastatin, FDA-approved Lescol as an anti-cholesterolemic drug, had a higher effect [179]. It modulates actin dynamics, resulting in the phosphorylation of Hippo effectors, a decrease in their activity and target genes’ expression, as well as in cell viability [165,167]. Fluvastatin also had an anti-CSC effect in breast cancer and promoted effects of chemotherapy drugs. Furthermore, Simvastatin was found to inhibit β-catenin expression and YAP activity, as well as the expression of target genes of both in GC cells. This resulted in a decrease in cell proliferation, migration, invasion, and apoptosis induction in vitro. Simvastatin effects passed through RhoA inhibition [166].

*Metformin*: HEK293 cells’ treatment with Metformin induces the AMPK pathway, which phosphorylates the Hippo adaptor protein AMOTL1, thus stabilising it. This facilitates and induces YAP phosphorylation by Hippo kinases, thus inhibiting the effectors [67]. Metformin is FDA-approved as an antidiabetic drug for type 2 diabetes [180]. In addition, Metformin has been shown to have anti-tumorigenic effects and, especially, anti-CSC capacities in GC. The authors demonstrate how Hippo modulation by Metformin targets EMT and CSC properties both in vitro and in vivo [168].

*Pazopanib*: Pazopanib is another drug that can be interesting for YAP/TAZ-TEAD targeting in GC. This FDA-approved drug, Votrient, for the treatment of Advanced Renal cell cancer and Advanced Soft tissue sarcoma, can also be repositioned for the targeting of YAP/TAZ-TEAD signalling in GC [181]. Known for its anti-VEGFR and PDGFR activities, Oku et al. demonstrate its role in YAP/TAZ-TEAD inhibition by driving the effectors towards degradation by the ubiquitin-proteasome system [165], through an effect on Hippo kinases. Pazopanib affects breast cancer cell and CSCs’ growth and survival as well as potentiates chemotherapies.

*SHAP*: In relation with the inhibitory effect of PP2A subunit STRN3 on Hippo kinases and its YAP-activation promoting effect described above [142], a highly selective peptide inhibitor was developed as targeted approach. The synthetised STRN3-derived Hippo-activating peptide SHAP was able to alter STRN3-PP2Aa interaction, restore MST1/2 phosphorylation capacity, and kinase the core tumour suppressor effect. Strong inhibition of YAP target genes was noted in the presence of SHAP, and this was associated with decreased cell viability, colony formation, and tumour cell growth in GC models [142].

*Leukaemia Inhibitory Factor (LIF)*: LIF, an interleukin 6-family cytokine was shown to activate MST/LATS and induce the phosphorylation of YAP/TAZ in GC, resulting in the inhibition of GC cells’ CSC and tumorigenic properties [43]. In addition, it blocks CSC-mediated drug efflux mechanisms and sensitises CSCs to chemotherapeutical drugs. The use of XMU-MP-1 MST1/2 inhibitor confirmed the role of the hippo pathway in LIF-induced effects. LIFR was found to be under-expressed in GC compared to non-tumoral tissues, and this was associated to low prognosis in patients, especially in diffuse-type GC. In addition, gene co-expression analysis using KMplotter brought to light the fact that patients having a low expression of LIFR, as well as high co-expression of oncogenic members of the Hippo pathway YAP/TAZ and target genes, had less survival chances compared to high-LIFR-expressing patients [43]. This proved the protective effect of LIF/LIFR signalling against Hippo effectors induction in GC. Similarly, Chen et al. demonstrated an anti-metastatic effect of LIF receptor (LIFR) in breast cancer, through the activation of hippo kinases and of adaptor protein Scribble [42]. In contradiction with this anti-metastatic effect of LIF/LIFR through the Hippo pathway, LIF was recently found to promote GC cell proliferation, invasion, and migration through the Hippo-YAP pathway [182]. Though the same cell models were not used, this demonstrated the complexity of this disease and this cell signalisation pathway.

## 4. Hippo Pathway-Aiming Strategies, Not Tested in GC

Several other strategies have been tested over the years in other models apart from GC, which could be of real interest in GC therapy. These are resumed below, and, again, involve strategies targeting the downstream oncogenic part of the pathway directly and those stimulating or aiming the upper kinase signalisation.

### 4.1. Downstream Strategies: Targetting Oncogenic YAP/TAZ-TEAD Signaling

Digitoxin, a cardiac glycoside, was identified as having an affinity with WW domains and inhibiting their interactions [183]. YAP/TAZ have WW domains, which can be targeted, and to this purpose, Sudol et al. constructed a structural model of YAP-digitoxin interaction that confirmed the binding and the interest of digitoxin in YAP-TEAD targeting [16]. Digitoxin, normally used for the treatment of cardiac arrythmias, has been shown to present anti-cancer properties [16,184,185,186]. It is able to induce apoptosis in NSCLC [16,187] and renal adenocarcinoma cells [187]. Studies show that it is able to inhibit HIF-1, which has been shown to interact with TAZ and to be associated with EMT [188]. Its HIF-1 inhibition capacity gives it anti-tumorigenic properties in in vivo mouse models of prostate cancer cells subcutaneous xenograft [188].

Furthermore, high-throughput screening has allowed the identification of C108, a chemical compound capable of decreasing YAP-dependent transcription activity. C108 decreases YAP protein expression, without changing its mRNA expression nor its phosphorylation [189]. This regulation of YAP does not involve activation of Hippo kinases nor a decrease in YAP-TEAD interaction, but is dependent on the ubiquitination and proteasomal degradation of YAP. Functionally, C108 decreases cell proliferation, migration, and tumour growth in vivo, while apoptotic-markercleaved PARP is found to increase. Effects were shown on melanoma and lung adenocarcinoma models, but could be tested in GC [189].

Among the different strategies are many TEAD-binding molecules having the capacity of altering YAP/TAZ-TEAD interaction. Indeed, structural understanding of YAP/TAZ binding domains (Y/TBD) of TEADs has allowed the search and/or design of molecules inhibiting YAP/TAZ-TEAD interaction. A central pocket has been discovered in TEAD’s Y/TBD, giving hope for its targeting by small molecule inhibitors.

Pobbati et al. show that a non-steroidal anti-inflammatory drug, Flufemanic acid, can interact with TEAD2 at the level of this pocket and, in so doing, inhibits YAP-TEAD interaction and transcriptional activity, as well as the consequent effect on cell proliferation and migration in HEK293 Human embryonic kidney cells [190]. Nevertheless, high concentrations of the molecule are needed for its effect take place [190]. Moreover, with the aim of improving its effect, Bum-Erdene et al. used the Flufemanic acid structure as inspiration to design a small molecule, TED-347, capable of forming a covalent bond with a conserved cysteine in the YBD pocket [191]. This covalent bond decreases YAP-TEAD4 binding affinity and irreversibly decreases YAP-TEAD transcriptional activity in a dose-dependent manner and glioblastoma cells’ viability [191]. Structural studies allowing proper understanding of YAP/TAZ-TEAD interaction has permitted the discovery, development, and synthesis of a multitude of YAP/TAZ-TEAD interaction inhibitors. Compound 9 was designed following hexT21-A56 peptoid discovery and is able to inhibit YAP-TEAD4 interaction, thus decreasing YAP target gene *CTGF* expression in MST1/2 inhibited HEK293 Human embryonic kidney cells [192]. Compound 19, an analogue of Kojic acid, also inhibits YAP-TEAD4 interaction and activity by binding TEAD [193]. Moreover, Kurppa et al. developed MYF-01-37, a small molecule able to covalently bind to cysteine 380 and 359 in TEAD2 and TEAD1, respectively [84]. YAP-TEAD interaction is inhibited by MYF-01-37 resulting in decrease in their target genes expression in PC-9 lung adenocarcinoma cells.

Small molecules screening using a novel ultra-bright NanoLuc biosensor, allowing the quantification of protein–protein interactions in living cells, allowed the identification of Celastrol as YAP/TAZ-TEAD inhibitor [194]. When tested on breast and lung cancer cell lines, Celastrol is able to decrease TEAD target genes expression, resulting in lower cell proliferation, survival, and migration, in fewer cells with CSC characteristics. Another strategy aiming at the disruption of YAP-TEAD interactions was proposed by Zhou et al., who engineered a YAP-like peptide that can fix itself on TEAD and occupy the YAP-binding region. This increases YAP-TEAD interaction disruption and seems to contribute to decreased tumour growth in HCC [194]. This works since this engineered peptide has a much better affinity for TEAD and competes with endogenous YAP.

Furthermore, studies show that cystine-dense peptides (CDP) have the capacity of altering protein–protein interactions and could be an interesting clue in the search for YAP/TEAD-targeting molecules. Crook et al. identify TB1G2 as a disulfide-stabilised CDP, presenting the capacity of interrupting YAP-TEAD interactions [195]. However, tests in HeLa cells show that is not able to penetrate cells, so further work is needed to try to make it cell permeable, to test its functions biologically [195]. Nevertheless, the authors present a mammalian platform capable of screening CDPs of interest in the desired protein–protein interaction. Using the Bristol University Docking engine (BUDE), allowing the screening of over 8 million drug-like compounds, Smith et al. identified, in silico, CPD3 as a novel TEAD interactor and inhibitor [196]. Experiments on HeLa cells confirmed CPD3′s effect as inhibitor of TEAD-dependent target gene induction, cell proliferation, and migration [196]. CPD3 protein–protein interaction inhibitor represses YAP activation of all TEAD-family members. This could be a problem, since TEADs are related to diverse physiological properties and dissecting these could allow the identification of even more targeting drugs. Sturbaut et al. synthesized pyrazoles, which are able to bind TEAD and, among which, ethyl 1-(4-aminobutyl)-3-(3,4-dichlorophenyl)-1H-pyrazole-4-carboxylate (named Compound 6 by the authors) is able to inhibit TEAD-induced target genes expression as well as cell proliferation [197].

Finally, virtual structure analysis and screening led to the discovery of DC-TEADin02, a vinylsulfonamide derivative, as a TEAD auto-palmitoylation inhibitor [198]. This inhibitor has a low effect on YAP-TEAD interaction but acts on TEAD palmitoylation, leading to decreased TEAD transcriptional activity and low target genes expression in HCT116 colon cells and HEK293 human embryonic kidney cells. This shows another TEAD-activity related to the targeting of hippo signalling, without affecting YAP-TEAD interaction [198].

### 4.2. Upstream Strategies: Hippo Kinases and/or Side-Pathways Stimulation

Melatonin binding to its GPCRs Melatonin receptor 1 and 2 (MT1 and MT2) suppresses TGF-β1-induced lung fibroblasts proliferation, through an effect on YAP activity [199]. Melatonin decreases YAP nuclear translocation and inhibits TGF-β-promoted cell migration and proliferation, thus repressing lung fibrosis. Furthermore, Dasatinib, a small inhibitor of SRC-family protein kinases, also represents an interesting strategy. It has been FDA-approved as Sprycel for use in the treatment of Acute lymphoblastic leukaemia and Chronic myelogenous leukaemia [200], and has been shown to inhibit nuclear localization and activity of YAP/TAZ in breast cancer cells as well as affect the viability of the latter [165]. Like Fluvastatin, Dasatinib is able to change actin dynamics and, thus, induce YAP/TAZ phosphorylation by hippo kinases. It also sensitizes breast cancer cells to doxorubicin and paclitaxel chemotherapies, and the effect is extended on breast CSCs [165,167]. The RAF/MEK/ERK pathway, having a role in YAP control as described above, can be targeted by Trametinib, a MEK1/2 inhibitor, and FR180204 inhibiting ERK, which leads to a decrease in YAP protein expression, but not that of its mRNA [65]. Furthermore, a decrease in transcriptional activity of YAP-TEAD is noted in lung cancer cells. Basu et al. identify C19 compound, a small molecule, as a potential inhibitor of the Hippo pathway, but of WNT and TGF-β-associated pathways too, which are also found to be regulated by Hippo, as described above [48,201]. C19 is able to induce degradation of TAZ, by acting on GSK3-β and AMPK and by activating Hippo kinases MST/LATS. Phosphorylation cascade results in inhibition of cell proliferation, migration, and doxorubicin resistance, as well as the anti-tumour effect in vivo in melanoma and/or breast cancer models. C19 is presented as a molecule inhibiting several EMT-inducing pathways [201].

Hippo oncogenic effectors YAP/TAZ targeting can, thus, be achieved by different strategies involving the Hippo members or not, and a pathway’s crosstalk is an important aspect of this signalling pathway. Despite its interest and potential relevance in clinics, there is, for now, no approved Hippo-pathway-targeting strategy in clinics. Several clinical trials are, nevertheless, in progress for the inhibition of TEAD-driven transcription in cancers [202]. It should, however, be underlined that the Hippo signalling importance and its role in physiology makes it complicated to carry out systemic inhibition or kinases activation as a therapeutical strategy. Specific GC cells or CSC markers are, thus, important to try to develop cell-targeted therapies and bypass systemic side-effects.

## 5. Conclusions

The Hippo pathway is a crucial pathway in physiology, and its dysregulation is essential in some pathologies. It has a major responsibility in the gastric carcinogenesis process, since the early steps of the disease. Different signalling pathways such as WNT, AMPK, JNK, MAPK-c-Jun, and TGF-β-associated ones are also involved in Hippo-induced carcinogenesis and/or regulation. The Hippo pathway effector core YAP/TAZ-TEAD is overactivated in most GC cases, where it controls CSC tumorigenic and invasive properties. Targeting Hippo signalling is of utmost importance in GC therapy. Molecular data indicate that upregulation of YAP/TAZ-TEAD and their target genes in GC is always associated with aggressiveness of the disease and bad prognosis in patients. This is true, particularly, in SRCC, which is particularly aggressive and for which there is currently an urgent need for new therapies. Many strategies are being tested, either targeting the downstream oncogenic effectors or stimulating the upstream tumour-suppressor kinase core. Structural knowledge about protein–protein interactions has given rise to new hopes for the targeting of YAP/TAZ-TEAD interactions, with the identification and design of small molecules capable of binding TEAD and inhibiting oncogenic bindings. Nevertheless, there is still much to clarify to be able to propose targeted therapies aiming at repressing YAP/TAZ tumorigenic and metastatic effects in GC. With the goal at term of developing combinatorial therapies allowing GC and gastric CSCs’ targeting, thus limiting relapse and resistance, for which the Hippo pathway is greatly responsible.

## Figures and Tables

**Figure 1 cancers-14-02282-f001:**
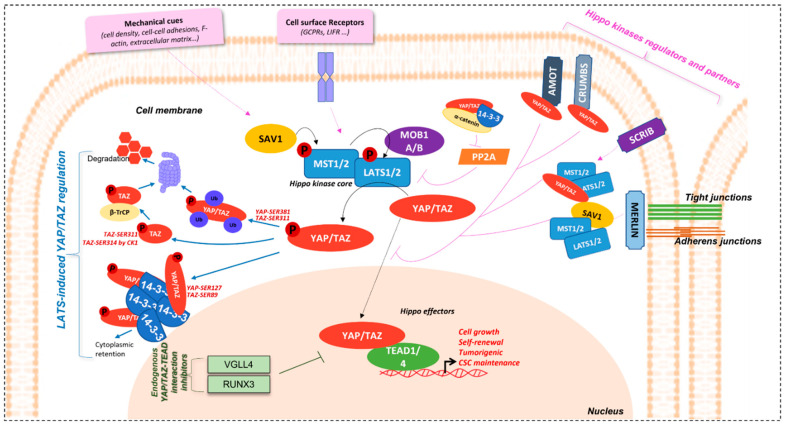
Hippo/YAP/TAZ-TEAD signalling pathway. Schematical representation of Hippo signalisation network. Regulation of YAP/TAZ effectors by LATS are represented by the blue arrows, endogenous YAP/TAZ-TEAD inhibitors in green, and Hippo kinases regulators and partners in pink.

**Figure 2 cancers-14-02282-f002:**
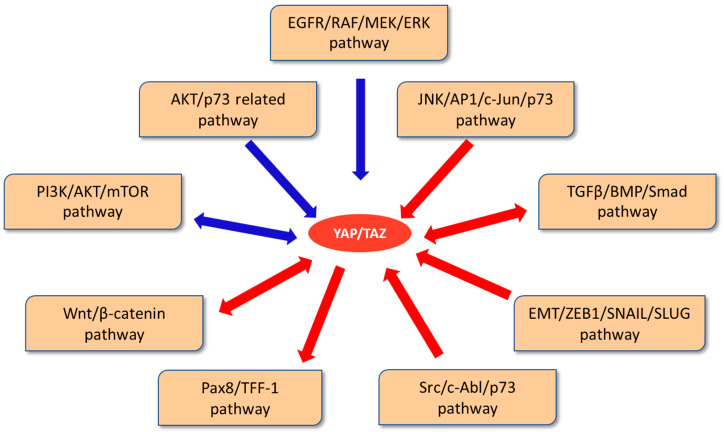
Crosstalk of YAP/TAZ with other signalling pathways than Hippo signalisation. YAP/TAZ negative regulation is represented by the blue arrows and positive regulation by the red ones.

**Figure 3 cancers-14-02282-f003:**
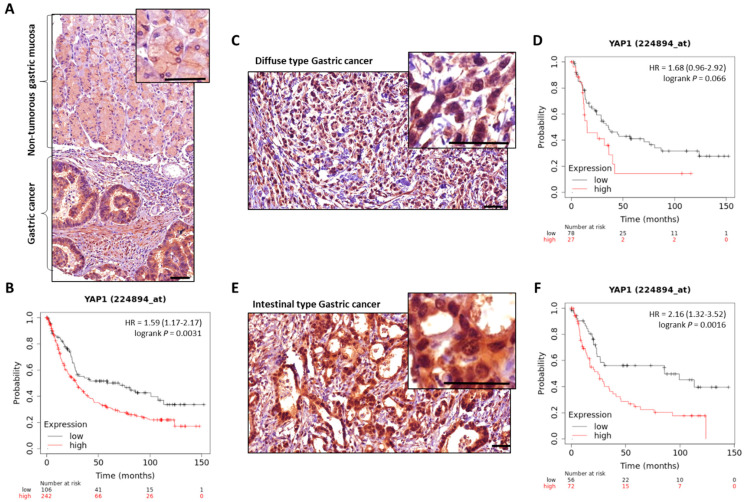
YAP expression in GC compared to non-tumorous gastric mucosa. Representative images of total-YAP immunostaining in GC patients’ tissues from our experiments. (**A**), non-tumorous gastric mucosa and adjacent GC from the same patient. (**C**,**E**) are representative images of diffuse and intestinal types GC cases, respectively; magnifications highlight YAP subcellular localisation, with nuclear accumulation in GC. Scale bars, 50 µm. KMplot™ analysis using KMplotter software (Kaplan-Meier plotter) [117] showing bad prognosis of patients with GC all subtypes included (**B**), diffuse (**D**), and intestinal (**F**) subtypes, when YAP is highly expressed.

**Table 1 cancers-14-02282-t001:** Summary of YAP/TAZ expression, activation and regulation in GC.

Expression Levels of YAP/TAZ	Regulation	Reference
Overexpression of YAP	Increase in pro-proliferation and pro-survival genes	[106]
	Upregulated through RUNX3 inactivation in GC	[31]
	Induced by METTL3 found highly expressed in GC	[139]
	Regulation by fixation of lncRNA *RP11-323N12.5* on its promoter	[145]
	Induced by HER4 and increases EMT, GC cells proliferation, and HER2-therapy resistance	[151]
Overactivation of YAP	Activated by PAR1 through inhibition of LATS Upregulation of stem-like properties	[126]
	Interacts with AMOTL1 to promote its nuclear translocation and activity	[140]
	Activation through MAPK-c-Jun pathway	[141]
	Inhibition is decreased through PP2A- inhibition of MST1/2	[142]
Overexpression of TAZ	Co-localisation with ZEB1 EMT transcription factor CSC tumorigenic properties	[107]
	Upregulation by MiR-125a-5p, leading to stimulation of genes involved in cell survival, EMT, invasion, and tumour growth	[87]
	Highly expressed in SRCC poorly undifferentiated GC	[153]
YAP/TAZ overexpression in CSCs and residual cells after chemotherapy-treatment	Overexpression of associated target genes	[125]

**Table 2 cancers-14-02282-t002:** Recapitulation of potential anti-GC molecules targeting the Hippo pathway.

Strategies	Molecules	Mechanism	Reference
Targeting oncogenic YAP/TAZ-TEAD signalling	*RUNX3*	YAP-TEAD interaction competitor	[31,151,152]
	*VGLL4*	YAP-TEAD interaction competitor	[30,113,153]
	*Super-TDU*	YAP-TEAD interaction competitor	[113]
	*Verteporfin*	YAP-TEAD interaction competitor Targets FAT1 and Survivin Induces 14-3-3 proteins PDT and induces cell death through singlet oxygen production	[121,149,154,155,156,157,158]
	*Peptide 17*	YAP-TEAD interaction inhibitor, Targets N6-methyladenosine (m6A)’s methyltransferase 3	[135,159,160]
	*WZ35, a Curcumin analogue*	Cell death through increase in cellular ROS level	[161,162]
Hippo kinases and/or side-pathways stimulation	*Dobutamine*	Recruits YAP to the cytoplasm through PKA signalling	[163]
	*Compound F10*	MST1/2 activation through cytoskeletal alteration	[164]
	*Statins (Lovastatin, Fluvastatin, Simvastatin)*	Modulates actin dynamics and activate Hippo kinases Inhibit β-catenin expression and YAP activity	[165,166,167]
	*Metformin*	Induces AMPK, which stabilizes AMOTL1 and induces Hippo kinases	[67,168]
	*Pazopanib*	Promotes effectors degradation by the ubiquitin-proteasome system	[165]
	*SHAP*	Alters STRN3-PP2Aa interaction and restores MST1/2 activity	[142]
	*Leukaemia Inhibitory Factor*	Induces LATS1/2 phosphorylation by MST1/2 and through Scribble activation in some cases	[42,43]

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
