# Peer review of "Hippo in Gastric Cancer: From Signalling to Therapy"

_cancers, 2022, doi:10.3390/cancers14092282_

Round 1

Reviewer 1 Report

Cancers-1653528

Seebeevassen et al.

Hippo in gastric cancer: from signaling to therapy

This is a comprehensive review of Yap/Taz via HIPPO signaling in gastric cancer.  In general, the review is quite long and rather unfocused, particularly because it includes a summary of HIPPO signaling in numerous other cancers thus diluting the overall focus on gastric cancer.  The main concern is that only a few recent 2021-2022 references are cited out of 80 or so written on this topic in that time frame.  It seems that thoughtful integration of the older literature with everything newly discovered would be most helpful-since the field is moving rapidly.  The section on therapeutics should also include pitfalls about using drugs to target a pathway utilized overall by every cell/tissue in the body to regulate size and to ensure homeostasis.  Specific comment/concerns are below.

Abstract:

In the simple summary and abstract, which will be read by everyone, it might be helpful to have someone edit the English.  Some words are not used properly in the context of the paragraph or the intended meaning therein should be defined in this section.

Many statements throughout the document need to be referenced.

Figure 1.  Suggest not calling the pathway an “oncogenic pathway” or the nuclear transcriptional machinery “oncogenic effectors”.  Numerous scientific articles have shown considerable amounts of active Yap in the nucleus of normal parietal cells, where it likely regulates homeostasis rather than being oncogenic.  The pathway shown is the signaling pathway, not necessarily an oncogenic signaling pathway.

If the review article is focused on gastric cancer, suggest to delete lines 252-364.  If gastric cancer pathogenesis uses paradigms from other cancers, the information might be woven into the discussion on gastric cancer.

Figure 3 a and b must be improved.  The “normal” gastric mucosa should be shown from surface through the base of glands and marked accordingly.  The gland base is usually shown at the bottom of the image and surface/pit cells at the top of the image.  It should be denoted whether the staining is for total YAP or activated YAP and high mags should be presented to show which cells have nuclear localization and which do not.  The type of gastric cancer is denoted but tissues are really over stained so the localization is not discernable.  Reference to the paper should be included in the legend.

Line 375, it is unclear how transcriptomic analysis would indicate the activation status of YAP, please revise, edit, or reference.

Discussion about Yap/Taz in gastric cancer, per se, might benefit from summary figures that integrate written materials concerning the expression levels, regulation, and therapeutics.

New references from the literature Jan 01, 2021 through present are lacking.  Thus, the new literature on this topic must be incorporated with the older papers in the review.  What has changed in the field in the past year or so-and how does this new information change potential therapeutic targets?

Considering that normal stomach mucosa expresses Yap1, Taz, TEADs and all of the upstream effectors, it seems that any discussion of therapeutic inhibition of the pathway be accompanied with a discussion about the ramifications of blocking the pathway on normal tissue homeostasis.  For systemic treatment, this comment also stands for other tissues that use this pathway, which is likely all organ systems. Additionally, how about combination therapies since it is highly unlikely that blocking HIPPO signaling alone in gastric cancer would be curative.

Author Response

Reviewer 1

This is a comprehensive review of Yap/Taz via HIPPO signaling in gastric cancer.  In general, the review is quite long and rather unfocused, particularly because it includes a summary of HIPPO signaling in numerous other cancers thus diluting the overall focus on gastric cancer.  The main concern is that only a few recent 2021-2022 references are cited out of 80 or so written on this topic in that time frame.  It seems that thoughtful integration of the older literature with everything newly discovered would be most helpful-since the field is moving rapidly.  The section on therapeutics should also include pitfalls about using drugs to target a pathway utilized overall by every cell/tissue in the body to regulate size and to ensure homeostasis.  Specific comment/concerns are below.

Abstract:

In the simple summary and abstract, which will be read by everyone, it might be helpful to have someone edit the English.  Some words are not used properly in the context of the paragraph or the intended meaning therein should be defined in this section.

Thank you for your comment, the article was read and revised by a native English speaker.

- Many statements throughout the document need to be referenced.

Figure 1.  Suggest not calling the pathway an “oncogenic pathway” or the nuclear transcriptional machinery “oncogenic effectors”.  Numerous scientific articles have shown considerable amounts of active Yap in the nucleus of normal parietal cells, where it likely regulates homeostasis rather than being oncogenic.  The pathway shown is the signaling pathway, not necessarily an oncogenic signaling pathway.

- We agree with your remark about the oncogenic word and thus modified the figure by removing “oncogenic” as well as “tumour suppressor” for the kinases. The figure legend was also modified by removing “oncogenic”.

If the review article is focused on gastric cancer, suggest to delete lines 252-364.  If gastric cancer pathogenesis uses paradigms from other cancers, the information might be woven into the discussion on gastric cancer.

- We think that it is important to talk about what is known in other cancers since Hippo pathway is a very complex and important pathway and the original studies demonstrating its pro-oncogenic role were not done in gastric cancer only and talking about them is important. The two other reviewers find this part good and clear. Because it was very difficult to modify a section accepted by the other reviewers, we have thus left it this way. We hope you will understand and that it will suit you.

Figure 3 a and b must be improved.  The “normal” gastric mucosa should be shown from surface through the base of glands and marked accordingly.  The gland base is usually shown at the bottom of the image and surface/pit cells at the top of the image.  It should be denoted whether the staining is for total YAP or activated YAP and high mags should be presented to show which cells have nuclear localization and which do not.  The type of gastric cancer is denoted but tissues are really over stained so the localization is not discernable.  Reference to the paper should be included in the legend.

- The figure orientation has been improved to change the gastric mucosa as it should normally be. These are original pictures from our lab showing YAP expression which is more marked in the cancer tissues and nuclear in both histological subtypes of GC. “total-YAP” and “from our experiments” were added to the figure legend to clear this out. In panel A, the image show the non-tumorous gastric mucosa with gastric cancer invading the submucosa and the muscularis layers. In Panel C and E are shown images of GC of diffuse type (C) and intestinal type (E). The image in E of the intestinal type GC was replaced by another one with a higher definition as requested. Magnifications have been added to show the subcellular localisation of YAP in normal gastric mucosa and its nuclear accumulation in tumors, as requested. The figure has been reorganised and replaced in the revised version of the manuscript.

Line 375, it is unclear how transcriptomic analysis would indicate the activation status of YAP, please revise, edit, or reference.

- Increased transcriptomic expression of YAP/TAZ target genes reflect YAP/TAZ activation. This is explained in the reference [106] Molina, CMGH 2020 cited there.

Discussion about Yap/Taz in gastric cancer, per se, might benefit from summary figures that integrate written materials concerning the expression levels, regulation, and therapeutics.

- In order to address your comment, a new table was added line 652 to resume YAP/TAZ expression levels and regulation in GC. The therapeutics part is summarised in table 2.  

New references from the literature Jan 01, 2021 through present are lacking.  Thus, the new literature on this topic must be incorporated with the older papers in the review.  What has changed in the field in the past year or so-and how does this new information change potential therapeutic targets?

- New reference from Jan 2021 was added line 1172 “In contradiction with this anti-metastatic effect of LIF/LIFR through the Hippo pathway, LIF was recently found to promote GC cell proliferation, invasion and migration through the Hippo-YAP pathway [182]. Though not the same cell models were used, this demonstrated the complexity of this disease and this cell signalisation pathway.”

Considering that normal stomach mucosa expresses Yap1, Taz, TEADs and all of the upstream effectors, it seems that any discussion of therapeutic inhibition of the pathway be accompanied with a discussion about the ramifications of blocking the pathway on normal tissue homeostasis.  For systemic treatment, this comment also stands for other tissues that use this pathway, which is likely all organ systems. Additionally, how about combination therapies since it is highly unlikely that blocking HIPPO signaling alone in gastric cancer would be curative.

- Thanks for your comment. Though we decided to focus on the targeting of Hippo through upstream Hippo kinases stimulation or downstream oncogenic effectors inhibition, combination therapy notion was mentioned in the conclusion. Concerning role of Hippo in physiology and impact of its inhibition or activation in normal cells, 2 sentences were added line 1001 “It should however be underlined that Hippo signalling importance and role in physi-ology makes it complicated to carry out systemic inhibition or kinases activation as therapeutical strategy. Specific GC cells or CSC markers are thus important to try to develop cell-targeted therapies and bypass systemic side-effects.”

Reviewer 2 Report

i congratulate the authors in a thorough and well organized review on the Hippo signaling pathway. i would like to challenge them using their own title....therapy...can the authors review any clinical data that support that this is a promising pathway to target for gastric or other gastrointestinal malignancies? that would really make the review interesting 

Author Response

Reviewer 2

i congratulate the authors in a thorough and well organized review on the Hippo signaling pathway. i would like to challenge them using their own title....therapy...can the authors review any clinical data that support that this is a promising pathway to target for gastric or other gastrointestinal malignancies? that would really make the review interesting

- Thanks a lot for your interesting comment. We have found and added this line 998 : “Despite its interest and potential relevance in clinics, there is for now no approved Hippo pathway-targeting strategy in clinics. Several clinical trials are nevertheless in progress for the inhibition if TEAD-driven transcription in cancers [202].”

Reviewer 3 Report

The review proposed by Seeneevassen is very interesting and goes deeper in the HIPPO pathway.

I suggest to summarize paragraph about HIPPO pathway description it results too long, even if is interesting.

Is fig.3 part of your experiments? if not please cite the origin.

I suggest also to shorten the following part about HIPPO in tumorigenesis and GC.  this is too detailed and readers can get lost

Author Response

Reviewer 3

The review proposed by Seeneevassen is very interesting and goes deeper in the HIPPO pathway.

I suggest to summarize paragraph about HIPPO pathway description it results too long, even if is interesting.

- Thanks for your comment and suggestion. This paragraph is complicated to shorten since it is important to understand the Hippo pathway and follow its dysregulation in cancer. Other reviewer found it clear. In addition, figure 1 is here to help summarise the whole regulation part which is of importance to understand hippo pathway dysregulation in pathology. Likewise, it is important to talk about the different regulators in this part since these are either dysregulated in cancer or can be used in the development of Hippo-targeted therapeutical strategies.   

Is fig.3 part of your experiments? if not please cite the origin.

- Yes it is part of our experiments. We have added “from our experiments” in the figure legend associated to this figure, line 387, to make this clearer.

I suggest also to shorten the following part about HIPPO in tumorigenesis and GC.  this is too detailed and readers can get lost

- Thanks for your comment, I understand your point. However, reviewer 2 found it clear and even mentioned to add some clinical knowledge. We decided to leave it as it is. We hope that it is good for you.

Round 2

Reviewer 1 Report

The revised version is improved significantly.

Reviewer 2 Report

thank you for the revised version